# A Direct Method for RT-PCR Detection of SARS-CoV-2 in Clinical Samples

**DOI:** 10.3390/healthcare9010037

**Published:** 2021-01-04

**Authors:** Sherif A. El-Kafrawy, Mai M. El-Daly, Ahmed M. Hassan, Reham M. Kaki, Adel M. Abuzenadah, Mohammad A. Kamal, Esam I. Azhar

**Affiliations:** 1Special Infectious Agents Unit, King Fahd Medical Research Center, King Abdulaziz University, Jeddah 21589, Saudi Arabia; saelkfrawy@kau.edu.sa (S.A.E.-K.); meldaly@kau.edu.sa (M.M.E.-D.); hmsahmed@kau.edu.sa (A.M.H.); aabuzenadah@kau.edu.sa (A.M.A.); 2Department of Medical Laboratory Technology, Faculty of Applied Medical Sciences, King Abdulaziz University, Jeddah 21589, Saudi Arabia; 3Department of Medicine, Department of Infectious Disease and Department of Infection Control and Environmental Health King Abdulaziz University Hospital, King Abdulaziz University, Jeddah 21589, Saudi Arabia; rmkaki@kau.edu.sa; 4King Fahd Medical Research Center, King Abdulaziz University, P. O. Box 80216, Jeddah 21589, Saudi Arabia; prof.ma.kamal@gmail.com; 5Enzymoics, 7 Peterlee Place, Novel Global Community Educational Foundation, Hebersham, NSW 2770, Australia

**Keywords:** SARS-CoV-2, direct RT-PCR, COVID-19, molecular detection

## Abstract

Introduction: the emergence of severe acute respiratory syndrome coronavirus 2 (SARS-CoV-2) has caused a global pandemic of acute respiratory disease (COVID-19). SARS-CoV-2 is a positive-strand RNA virus and its genomic characterization has played a vital role in the design of appropriate diagnostics tests. The current RT-PCR protocol for SARS-CoV-2 detects two regions of the viral genome, requiring RNA extraction and several hours. There is a need for fast, simple, and cost-effective detection strategies. Methods: we optimized a protocol for direct RT-PCR detection of SARS-CoV-2 without the need for nucleic acid extraction. Nasopharyngeal samples were diluted to 1:3 using diethyl pyrocarbonate (DEPC)-treated water. The diluted samples were incubated at 95 °C for 5 min in a thermal cycler, followed by a cooling step at 4 °C for 5 min. Samples then underwent reverse transcription real-time RT-PCR in the E and RdRp genes. Results: our direct detection protocol showed 100% concordance with the standard protocol with an average Ct value difference of 4.38 for the E region and 3.85 for the RdRp region. Conclusion: the direct PCR technique was found to be a reliable and sensitive method that can be used to reduce the time and cost of the assay by removing the need for RNA extraction. It enables the use of the assay in research, diagnostics, and screening for COVID-19 in regions with fewer economic resources, where supplies are more limited allowing for wider use for screening.

## 1. Introduction

Severe acute respiratory syndrome coronavirus 2 (SARS-CoV-2) was first reported in Wuhan, China and quickly spread globally. It causes a disease called coronavirus disease 2019 (COVID-19) that includes acute pneumonia [1,2]. Coronaviruses are large, enveloped, positive single-stranded RNA viruses of zoonotic origin that infect humans and a wide range of animals. The genome sizes range from 26 to 32 kb. There are four subfamilies of coronaviruses; alpha, beta, gamma, and delta. Alpha and beta originate from mammals, especially bats, while gamma and delta originate from birds and pigs. SARS-CoV-2 belongings to the B strain of beta-coronaviruses, and is closely related to SARS-CoV virus [3]. The four major structural genes encode the nucleocapsid protein (N), spike protein (S), membrane glycoprotein (M), and small membrane protein (SM). SARS-CoV-2 shows a 96% sequence similarity to the genome of a bat coronavirus [3]. 

High-volume testing together with case isolation and contact tracing is one of the most effective strategies to prevent the spread of the infection in the community [4]. One of the WHO recommendations for testing the clinical samples to confirm SARS-CoV-2 infection is the RT-PCR amplifications in two regions of the viral genome. The mostly used regions for detection is are the structural E region and the non-structural RdRp region. The emergence of the virus with the global spread of infection and the high demand for testing kits and reagents has led to a delay in development and supply of testing kits and reagents, which delays the screening of infected individuals and their contacts [4]. This has slowed public health authorities’ response to the global pandemic, especially in developing and low-income countries. The urgent and increasing need for reagents, supplies, and kits for testing has limited the availability, and led to poor logistical support for obtaining testing supplies which rendered suppliers unable to cope with demand.

In response to the shortage of nucleic acid extraction kits needed for real-time reverse transcription PCR (rtRT-PCR) assays, we developed a method to perform the molecular assay without the need for nucleic acid extraction. The adapted method was based on boiling the sample at 95 °C to deactivate the PCR inhibitors and liberate the viral nucleic acids. This method overcomes the shortage of nucleic acid extraction kits, reduces the cost, and reduces the time from sampling to results for SARS-CoV-2 detection by rtRT-PCR. The protocol was adapted from previous methods to detect other viruses in various sample types, including serum (for dengue virus and HCV) and respiratory samples [5,6,7,8]. In these protocols, nucleic acid extraction was replaced by heat or chemical inactivation followed by amplification of the target organism’s gene(s). 

There have been several reports of direct PCR to detect SARS-CoV-2 in clinical samples. The first [9] used heating at 70 °C for 10 min to pretreat the sample before RT-PCR amplification. The authors reported an average increase in the Ct values of 6, leading to a false negative rate of 12%. The second report [10] used direct rtRT-PCR for sputum and nasopharyngeal exudates that were spiked with a plasmid containing the SARS-CoV-2 N gene. They reported a lowest limit of detection (LoD) of 2 copies per reaction for spiked sputum and 20 copies per reaction for spiked nasal exudates. Two more reports have utilized the same approach for direct detection of SARS-CoV-2 without nucleic acid extraction, one study reported the heat treatment of respiratory samples followed by RT-PCR their best sensitivity was achieved when using 3 μL sample volume [11]. Their results showed that the direct RT-PCR can replace extraction-based SARS-CoV-2 diagnostics. The fourth study [12] of Ioanna et al. found that heat inactivation of the samples prior PCR reactions at 95 °C gave superior results than the incubation at 65 °C and also lead to virus inactivation for better biosafety.

We report a direct rtRT-PCR protocol to detect SARS-CoV-2 that is based on diluting the clinical samples, pretreating them with heat, and then amplifying the nucleic acids with rtRT-PCR. The protocol was validated on 130 clinical samples and showed 100% concordance with the standard extraction protocol.

## 2. Methods

### 2.1. SARS-CoV-2 Virus Titration

The procedure was performed in the BSL3 laboratory of the Special Infectious Agents Unit, King Fahd Medical Research Center, King Abdulaziz University. In this experiment, the virus stock of the third blind passage (accession number MT630423) was serially diluted within the range of 10^−1^ to 10^−12^. Working virus dilutions were added to the green monkey kidney VERO E6 cells in 96 wells plates with seven wells per dilution to calculate the tissue culture infectious dose 50% (TCID_50_) of the virus stock. The cell culture plate was incubated at 37 °C with 5% CO_2_, and the cells were observed daily until the cytopathic effect (CPE) was set. The virus titer was calculated to be 2.2 × 10^6^ pfu/mL.

### 2.2. Viral RNA Extraction

The extraction was performed using the Sample Preparation System Nucleic Acid Extraction Kit (Promega, Walldorf, Germany) on the Te-magS magnetic separator (Tecan, Crailsheim, Germany) with a 700 μL sample volume and an 80 µL elution volume. Sample processing was performed in a class II biosafety cabinet in a negatively pressured lab. 

### 2.3. Sample Preparation Optimization for Direct rtRT-PCR

Cell culture supernatants and nasopharyngeal swabs collected from suspected COVID-19 cases in viral transport media were boiled in a 0.2 mL 96-well plate sealed with an adhesive film to minimize carryover contamination and ensure biosafety. Samples were diluted at different ratios ranging from no dilution to 1:5 dilution in either DEPC-treated water or viral lysis buffer (AVL) (Qiagen, Hilden, Germany). The diluted samples were incubated at 95 °C for 5 or 10 min in a thermal cycler, followed by cooling at 4 °C for 5 min (Figure 1).

### 2.4. RT-PCR 

The reaction master mix consisted of 11 µL RT-PCR Premix and 4 µL of Primer/Probe Mix (PowerChek 2019-nCov Real-Time PCR Kit, Seoul, Korea) for each sample. To this, 5 μL of boiled sample or the RNA extract was added for a total reaction volume of 20 µL. As per WHO recommendations, two targets were separately amplified: the E gene of beta coronaviruses and the RdRp gene of the SARS-CoV-2 viral genome [13]. Both probes were labeled with FAM. The reaction mixture included an internal control for PCR inhibition with a Victoria (VIC)-labelled probe. The reaction’s thermal profile was 50 °C for 30 min, then 95 °C for 10 min, followed by 40 cycles of 95 °C for 15 s and 60 °C for 1 min.

The lowest LoD was estimated using serial 10-fold dilutions of SARS-CoV-2 isolate supernatants in viral transport medium. The LoD was estimated to be the highest dilution giving Ct values less than 37 in all replicates.

## 3. Results

The sample diluent was optimized by testing matched dilutions in DEPC-treated water and Buffer AVL. Buffer AVL completely inhibited the PCR reactions and did not yield positive results for any of the dilutions, so the remaining sample preparations used DEPC-treated water. The time for sample denaturation (95 °C) was evaluated at 5 and 10 min, with 10 min showing significantly higher Ct values and decreased sensitivity (Table 1). Therefore, a 5 min sample denaturation was used for subsequent experiments. Undiluted samples resulted in increased PCR inhibition and delayed amplification curves. The best results were obtained with 1:3 dilutions (10 μL sample and 30 μL DEPC-treated water) at 95 °C for 5 min, followed by 4 °C for 5 min.

The direct PCR assay showed 100% concordance with the standard rtRT-PCR technique. However, the Ct values for our direct rtRT-PCR technique were delayed relative to the standard technique by 2.71 cycles for the E gene and 1.32 cycles for the RdRp gene. This difference could be due to either slight reaction inhibition by the direct rtRT-PCR or the difference in volumes used. To evaluate the cause of this delay, we adjusted the volumes of the extracted RNA to match the input volume of the samples used in the direct assay. When compared to these diluted samples, the Ct delay for the direct protocol was largely eliminated (0.06 cycles for the E gene and 0.17 cycles for the RdRp gene). Figure 2 shows the variation in Ct values between the direct and extracted rtRT-PCR reactions before and after sample volume adjustment.

The LOD was calculated using a live SARS-CoV-2 cell culture supernatant from a clinical isolate that was 10-fold serially diluted in the range of 1.1 × 10^5^ to 1.1 × 10^−2^ pfu/mL. These dilutions were tested in triplicates using both direct and standard protocols to detect the E and RdRp genes. Table 1 shows the resulting average Ct values. The sensitivity of our direct PCR protocol was 1.1 pfu/mL, which is comparable to that of the standard RNA protocol (1.1 pfu/mL). The direct assay showed 100% concordance with the standard protocol down to 1.1 pfu/mL for both PCR targets (Table 1). At 1.1 × 10^1^ pfu/mL, both protocols failed to detect the RdRp gene, and the standard protocol detected the E gene with a Ct value of 38.44, but the direct protocol did not (Figure 3 and Figure 4). 

### Validation of the Direct RT-PCR Method Using Clinical Samples

The direct rtRT-PCR method was validated using 100 clinical nasopharyngeal samples that were sent to the Special Infectious Agents Unit for diagnostic purposes. Typically, the samples are analyzed using RNA extraction followed by rtRT-PCR with the same amplification reagents used for the direct method. The rtRT-PCR was then repeated in the same plate as the direct PCR samples to guarantee the same conditions for both assays. The standard rtRT-PCR protocol served as a reference to evaluate the performance of the direct PCR protocol. Of the 100 samples, 27 were negative using the standard rtRT-PCR, and 73 were positive with Ct values ranging from 14.33 to 35.1 (Figure 5). The direct protocol showed 100% concordance with the standard protocol with an average difference in Ct values of 4.38 for the E gene and 3.85 for the RdRp gene.

To evaluate the performance of the direct assay in clinical samples of low viral load we tested a set of 30 samples that showed Ct values higher than 31 in the standard protocol. All samples showed complete concordance when tested using the direct protocol, including 3 samples that showed high Ct values (35.2–36.9) in the E region and were negative when tested in the RdRp region.

## 4. Discussion

The need for a reliable, simple assay that identifies COVID-19 cases is in high demand to combat the spread of infection. Delays in obtaining extraction kits and the added time and cost of these kits led us to optimize and evaluate a direct rtRT-PCR protocol to detect SARS-CoV-2 in COVID-19 patients. To evaluate the lower LoD, we tested the assay’s performance in 10-fold serial dilutions of the tissue culture supernatant of a clinical isolate (accession number MT630423). The performance of the assay was comparable to the performance of the standard assay that includes viral RNA extraction. The LoD for both the standard and direct protocols was 1.1 pfu/mL. At this dilution, all replicates gave Ct values below 37; the next dilution (0.11 pfu/mL), resulted in no amplification with the direct protocol and detection of only the E gene with the standard protocol. 

Nucleic acid extraction is performed in PCR assays to remove PCR inhibitors and to liberate nucleic acids from the surrounding envelopes or viral proteins. To substitute for the nucleic acid extraction, samples were diluted to decrease the concentrations of PCR inhibitors that might be present in the samples. The dilution was performed in either DEPC-treated water or a viral lysis buffer (Buffer AVL) to deactivate proteases. The lysis buffer was found to inhibit the PCR reaction, failing to amplify any of the targets, including the internal control, possibly because it contains a chaotropic salt (guanidine isothiocyanate) that has the potential to deactivate the PCR enzymes. The samples were tested either without dilution or with a sample-to-diluent ratio that ranged from 1:1 to 1:5 in DEPC-treated water. Undiluted samples inhibited the PCR reaction, yielding delayed amplification and false negative results of samples with high Ct values. The best performance was achieved at a 1:3 dilution. This dilution seems to provide the optimum balance between input RNA and reducing the concentrations of PCR inhibitors. A similar observation was found by Smyrlaki et al. [12] where they reported that dilution of the clinical samples is essential to dilute the inhibitors that might be present in the samples. Heating the diluted specimen to 95 °C followed by quenching at 4 °C for 5 min was used to deactivate possible PCR inhibitors that might be present in the samples and to denature viral proteins. The duration of the heating step was tested at 5 and 10 min. While the 5 min denaturation provided a comparable LoD to the standard protocol, the 10-min denaturation resulted in much higher Ct values, possibly due to viral RNA degradation. Heating at 95 °C was reported by Smyrlski et al. [12] to help inactivate the nucleases and inhibitors in the clinical samples and was most importantly found to inactivate SARS-coV-2. Bruce et al. [11] also showed that heating is important for the detection of low viral load samples most probably since it denatured RNases/inhibitors of the enzymes and/or improved the accessibility of viral RNA through direct lysis of cells and virions.

The observed delay in the Ct value for the direct PCR relative to the standard protocol was found to be due to the difference in the input RNA between the two methods. The RNA extraction elutes an initial 700 μL of sample in 80 μL of elution buffer, resulting in an approximately 9-fold concentration of the sample RNA. In contrast, the direct PCR sample preparation results in a 4-fold dilution of the sample. Adjusting the input amount of the extracted RNA to match the volume of sample used for direct PCR almost abolished the Ct value delay. The optimized protocol was then validated in 100 respiratory samples that were also investigated using the standard protocol. The direct protocol showed complete concordance with the standard protocol with a negative result for the negative samples and an average delay in the Ct values of 4.38 for the E gene and 3.85 for the RdRp gene for positive samples. The performance of the assay proved also effective in detecting SARS-CoV2 in clinical samples with low viral load where it showed complete concordance with the extracted protocol in these samples.

Wee et al. [10] evaluated a direct rapid PCR test using synthetic SARS-CoV-2 RNA as a template spiked into negative sputum or nasopharyngeal aspirate. They used two approaches to detect the viral genome: one without extraction using the standard thermal cycling times and one with reduced denaturation, annealing, and extension times to decrease the time needed to obtain result. While their assay reported a LoD of 12 copies/mL for the direct PCR and 60 copies/mL for the rapid direct PCR, the assays were not evaluated using real patient samples. Another study by Alcoba-Florez et al. [9] described a fast method of SARS-CoV-2 detection by rtRT-PCR in preheated nasopharyngeal swabs. The authors evaluated the assay using nasopharyngeal samples from COVID-19 patients. Of the 41 positive samples, five were false negatives by the direct PCR compared to the standard protocol. 

Taken together, results from our optimized direct protocol reliably detected SARS-CoV-2 viral RNA in nasopharyngeal samples. The protocol can be adapted to other commercial or in-house amplification reagents after optimizing the amplification conditions. The assay also provides a time-saving and cost-effective alternative when nucleic extraction reagents are not available. 

## Figures and Tables

**Figure 1 healthcare-09-00037-f001:**
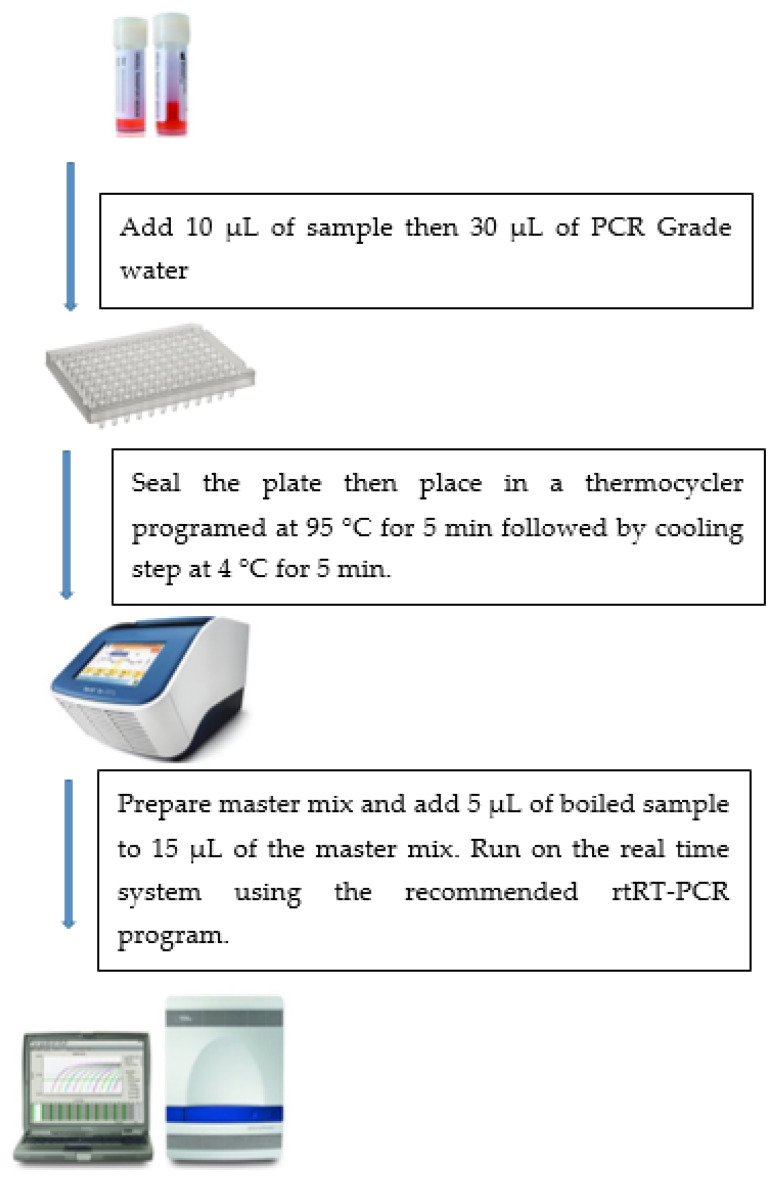
Schematic flow chart for the direct rtRT-PCR method.

**Figure 2 healthcare-09-00037-f002:**
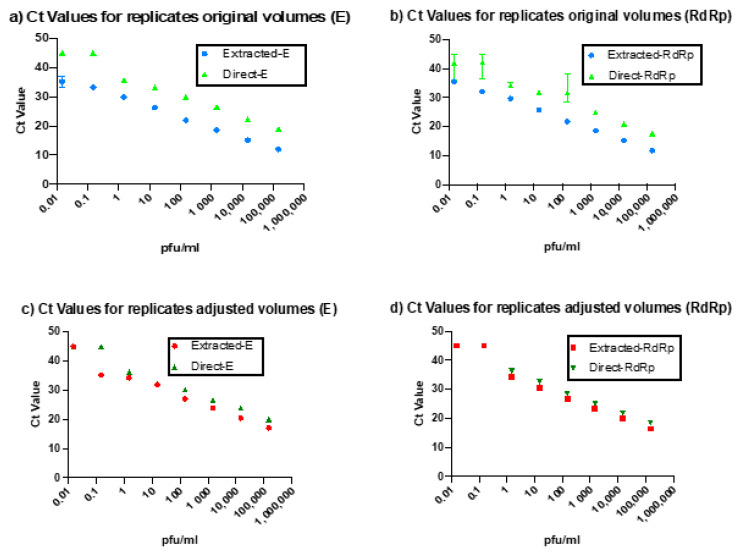
Comparison of Ct values between the extracted protocol and direct protocol using the original volumes of the extracted RNA in the E-region (**a**) and RdRp-region (**b**) compared to the extracted RNA after diluting the extracted RNA in the E-region (**c**) and the RdRp-region (**d**) to match the quantities used in the direct PCR.

**Figure 3 healthcare-09-00037-f003:**
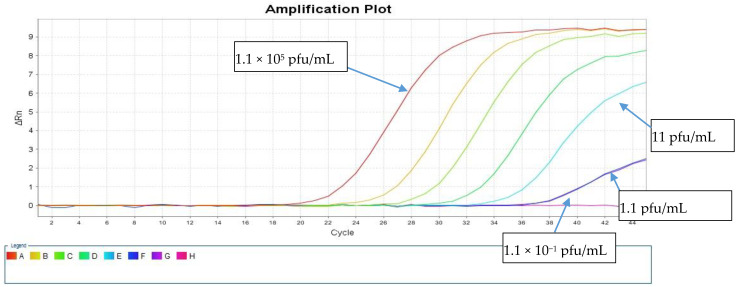
Amplification curves of the rtRT-PCR in the RdRp region using the virus isolates at serial 10-fold dilutions using the direct method.

**Figure 4 healthcare-09-00037-f004:**
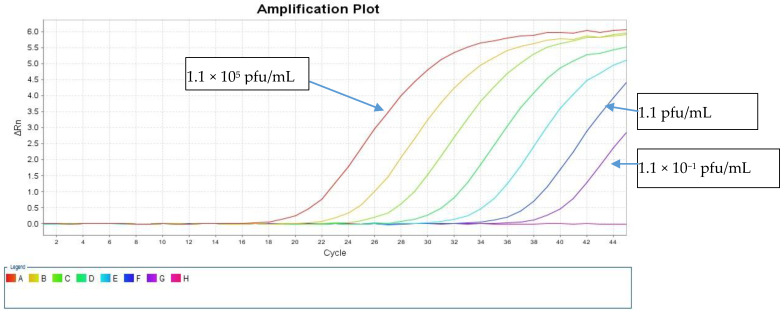
Amplification curves of the rtRT-PCR in the E region using the virus isolates at serial 10-fold dilutions using the direct method.

**Figure 5 healthcare-09-00037-f005:**
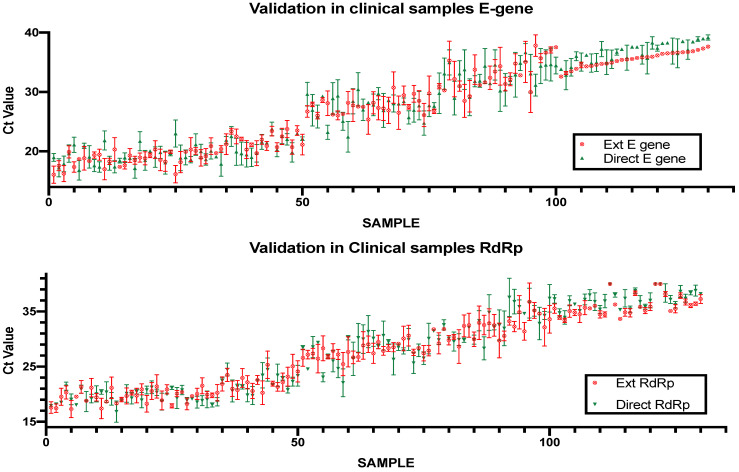
Evaluation of the direct rtRT-PCR (green triangles) using clinical samples compared to the standard extraction protocol (red crossed circles).

**Table 1 healthcare-09-00037-t001:** Ct values of culture supernatants used in the study for standard extraction protocol and for direct protocol at two denaturation times.

Virus Dilution	Virus Titer pfu/mL	Extracted RNA		Direct PCR 5 min Denaturation		Direct PCR 10 min Denaturation	
		E	RdRp	E	RdRp	E	RdRp
10^−1^	110,000	17.08	16.39	20.23	18.45	37.4	36.9
10^−2^	11,000	20.59	19.95	23.88	21.78	39.8	39.2
10^−3^	1100	23.91	23.43	26.75	24.96	ND	ND
10^−4^	110	27.16	26.86	30.37	28.52	ND	ND
10^−5^	11	31.94	30.76	32.35	32.30	ND	ND
10^−6^	1.1	34.08	34.28	36.28	36.20	ND	ND
10^−7^	0.11	38.44	ND	ND	ND	ND	ND
10^−8^	0.011	ND	ND	ND	ND	ND	ND

RNA: ribonucleic acid, PCR: polymerase chain reaction, RdRp: RNA dependent RNA polymerase, E: Envelope, ND: not detected.

## Data Availability

No new data were created or analyzed in this study. Data sharing is not applicable to this article.

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
