# Peer review of "A Direct Method for RT-PCR Detection of SARS-CoV-2 in Clinical Samples"

_healthcare, 2021, doi:10.3390/healthcare9010037_

Round 1

Reviewer 1 Report

The manuscript (ISSN 2227-9032) submitted by El-Kafrawy et al. describes a protocol that allows the direct detection of SARS-CoV-2 genome in nasopharyngeal swabs samples, without the need of nucleic acid extraction. This type of work is not new however, it is useful and important to have this type of approaches that allow to save time and costs, especially in low-income countries.

Nonetheless, and considering some other studies in the literature, do the authors know if this approach works in clinical samples with a low viral load?

Even So, the manuscript is well written and understandable, however it could be improved with more information in the introduction and in the discussion sections, since, beyond the cited reports, there are more studies that performed direct PCR to detect SARS-CoV-2 in clinical samples.

Other comments:

Page 5, line 155-156: The volumes of the extracted RNA were adjusted to match the input volume of the samples directly used. How did you do this? The initial volume of the extracted RNA used in the qPCR reactions was 5µL, so did you raise this volume?

Page 5, Table 1: The legend of the table should be self-explanatory, so the authors should mention that the Ct values presented to the extracted RNA are with the adjusted volumes.

Page 6, line 168-169: “The sensitivity of our direct PCR protocol was 1.1x10-1 fu/mL, which is comparable to that of the standard RNA protocol (1.1x10-1 pfu/mL).” According with table 1 the qPCR sensitivity is 1.1 pfu/mL and not 1.1x10-1 pfu/mL.

In the following lines the authors confirm that the both protocols failed to detect RsRp gene at 1.1x10-1 pfu/mL, so this value can not be pointed out as the sensitivity limit.

Page 7, Figure 2 and 3: Why does these figures appear after figure 4? Also, according with the presented information Gene E is the one with the highest sensitivity (1.1x10-1 pfu/mL) and not the RdRp gene, was there an exchange in the legend of the figures?

Page 9, line 235 and following: There are more studies available in the literature with similar approaches that could improve your work and help to enrich the manuscript.

Author Response

Reviewer 1:

Comments and Suggestions for Authors

The manuscript (ISSN 2227-9032) submitted by El-Kafrawy et al. describes a protocol that allows the direct detection of SARS-CoV-2 genome in nasopharyngeal swabs samples, without the need of nucleic acid extraction. This type of work is not new however, it is useful and important to have this type of approaches that allow to save time and costs, especially in low-income countries.

  1. Nonetheless, and considering some other studies in the literature, do the authors know if this approach works in clinical samples with a low viral load?

Response: The assay was tested further using cases with higher Ct values after the submission and the results are represented in the revised version.

  1. Even So, the manuscript is well written and understandable, however it could be improved with more information in the introduction and in the discussion sections, since, beyond the cited reports, there are more studies that performed direct PCR to detect SARS-CoV-2 in clinical samples.

Response: More studies were published, and these were discussed in the introduction and the discussion sections of the revised version of the manuscript.

Other comments:

  1. Page 5, line 155-156: The volumes of the extracted RNA were adjusted to match the input volume of the samples directly used. How did you do this? The initial volume of the extracted RNA used in the qPCR reactions was 5µL, so did you raise this volume?

Response: We thank the reviewer for this explanatory question, the input extracted RNA was diluted using DEPC treated water to match the input sample of the direct assay. As the extraction uses 700ml of sample and RNA is eluted in 80ml this results in a concentration of the input RNA by 8.75-fold. So, to match the input extracted RNA with the direct assay, extracted samples were diluted before RT-PCR and then subjected to RT-PCR reaction.

  1. Page 5, Table 1: The legend of the table should be self-explanatory, so the authors should mention that the Ct values presented to the extracted RNA are with the adjusted volumes.

Response: The Ct values presented in table 1 are made with unadjusted volumes in the extracted RNA columns.

  1. Page 6, line 168-169: “The sensitivity of our direct PCR protocol was 1.1x10-1 fu/mL, which is comparable to that of the standard RNA protocol (1.1x10-1 pfu/mL).” According with table 1 the qPCR sensitivity is 1.1 pfu/mL and not 1.1x10-1 pfu/mL.

In the following lines the authors confirm that the both protocols failed to detect RsRp gene at 1.1x10-1 pfu/mL, so this value can not be pointed out as the sensitivity limit.

Response: We apologize for this mistake, the statement was changed to “The sensitivity of our direct PCR protocol was 1.1 pfu/mL, which is comparable to that of the standard RNA protocol (1.1 pfu/mL)”.

  1. Page 7, Figure 2 and 3: Why does these figures appear after figure 4? Also, according with the presented information Gene E is the one with the highest sensitivity (1.1x10-1 pfu/mL) and not the RdRp gene, was there an exchange in the legend of the figures?

Response: We apologize for this formatting error, also the figure legends were exchanged and now they are in the correct order.

  1. Page 9, line 235 and following: There are more studies available in the literature with similar approaches that could improve your work and help to enrich the manuscript.

Response: These studies were reported in the introduction and discussed in the discussion section in the revised version of the manuscript.

Reviewer 2 Report

This work by El-Kafrawy and El-Daly, et al describes modifications to the SARS-CoV-2 diagnostic test that omit the RNA extraction step of the RT-PCR approach. As other groups have described, this is a technique that can simplify and streamline the traditional RT-PCR test, and also address reagent shortages that have posed a major problem during the COVID-19 pandemic.  This work builds upon previous preprints and published studies and describes refinements (dilution in water, 5 minutes vs 10 minutes of heat inactivation) that improves the sensitivity of previously published methods.

Major points

  • The authors should include more samples at higher CTs (where the method is likely to have difficulty), and/or discuss this limitation in their discussion.  While its true the approach has 100% concordance for the selection of clinical samples tested, the authors did not test any samples above 35.1, which is precisely where this approach will struggle based on previously published papers.  See Bruce et al (33006983) and Smyrlaki et al (32968075). The full data set used in Fig 5 should be included as a table as well, as the graph is difficult see the per-sample drop in sensitivity.  This table should also include data from the negative samples tested.

Minor points:

  • E should be mentioned as a structural protein in the introduction
  • There is a typo in Figure 4, the titles for Ct Values for replicates adjusted volumes E/(RdRp) has an extra “I” in adjusted.
  • In figure 5 it is difficult to see all the data points, due to the solid points and the fact some of the data overlaps. If the authors could make these data points partially transparent it would help to visualise the full data set. Ordering samples by CT would also help trends be more easily visualised- it is difficult to tell from this graph which green triangle goes with which red circle and see the difference between the methods. 
  • More detailed legends of Fig 2 and 3 would be helpful.
  • There are multiple studies verifying that direct RT-PCR can be successfully used to detect SARS-CoV-2 RNA in patient samples (refs mentioned above).  This work should be cited and discussed.

Author Response

Reviewer 2:

Comments and Suggestions for Authors

This work by El-Kafrawy and El-Daly, et al describes modifications to the SARS-CoV-2 diagnostic test that omit the RNA extraction step of the RT-PCR approach. As other groups have described, this is a technique that can simplify and streamline the traditional RT-PCR test, and also address reagent shortages that have posed a major problem during the COVID-19 pandemic. This work builds upon previous preprints and published studies and describes refinements (dilution in water, 5 minutes vs 10 minutes of heat inactivation) that improves the sensitivity of previously published methods.

Major points

  1. The authors should include more samples at higher CTs (where the method is likely to have difficulty), and/or discuss this limitation in their discussion. While its true the approach has 100% concordance for the selection of clinical samples tested, the authors did not test any samples above 35.1, which is precisely where this approach will struggle based on previously published papers.  See Bruce et al (33006983) and Smyrlaki et al (32968075).

Response: We thank the reviewer for this comment. In response to the reviewer comments we have tested 30 more samples with high Ct values. The results are presented in the results section figure 5 and discussed in the discussion section.

  1. The full data set used in Fig 5 should be included as a table as well, as the graph is difficult see the per-sample drop in sensitivity. This table should also include data from the negative samples tested.

Response: We thank the reviewer for this comment, but after addition of the extra tested samples the table will be very large. Instead, figure 5 in now modified by extending the length of the X-axis to overcome the crowding of the graph points and the red dots are replaced with red circles to show overlapping points for more clarity. If the reviewer insists on adding the table, we can still add it.

Minor points:

  1. E should be mentioned as a structural protein in the introduction

Response: the following statement was added to the introduction “One of the WHO recommendations for testing the clinical samples to confirm SARS-CoV-2 infection is the RT-PCR amplifications in two regions of the viral genome. The mostly used regions for detection is are the structural E region and the non-structural RdRp region.”

  1. There is a typo in Figure 4, the titles for Ct Values for replicates adjusted volumes E/(RdRp) has an extra “I” in adjusted.

Response: We apologize for this typing error; the figure titles were modified according to the reviewer comment.

  1. In figure 5 it is difficult to see all the data points, due to the solid points and the fact some of the data overlaps. If the authors could make these data points partially transparent it would help to visualise the full data set. Ordering samples by CT would also help trends be more easily visualised- it is difficult to tell from this graph which green triangle goes with which red circle and see the difference between the methods.

Response: We thank the reviewer for this comment, figure 5 is now modified by extending the length of the X-axis to overcome the crowding of the points and the red dots are replaced with red circles to show overlapping points for more clarity.

  1. More detailed legends of Fig 2 and 3 would be helpful.

Response: Figures 2 and 3 are now figures 3 and 4 their legend now read “Figure 3 Amplification curves of the rtRT-PCR in the RdRp region using the virus isolates at serial 10-fold dilutions using the direct method and Figure 4 Amplification curves of the rtRT-PCR in the E region using the virus isolates at serial 10-fold dilutions using the direct method”.

  1. There are multiple studies verifying that direct RT-PCR can be successfully used to detect SARS-CoV-2 RNA in patient samples (refs mentioned above). This work should be cited and discussed.

Response: The two studies mentioned above were described in the introduction section and their findings were discussed in the discussion section.

Round 2

Reviewer 1 Report

I considered the manuscript ready for publication, and don’t have any more comments or suggestion to add.